# A New Ammonia Kinetic Model in Ru-Catalyzed Steam-Reforming Reaction Containing $N_2$ in Natural Gas

**Chulmin Kim** , **Juhan Lee** **and Sangyong Lee** *

Department of Mechanical, Robotics and Energy Engineering, Dongguk University, Seoul 04620, Republic of Korea; smkcm@naver.com (C.K.); leejuh@dgu.ac.kr (J.L.)
* Correspondence: sangyonglee@dongguk.edu

**Abstract:** Hydrogen for building fuel cells is primarily produced by natural-gas steam-reforming reactions. Pipeline-transported natural gas in Europe and North America used to contain about 1% to 5% $N_2$, which reacts with $H_2$ in steam-reforming reactions to form $NH_3$. In the case of Ru, one of the catalysts used in natural-gas steam-reforming reactions, the activity of the $NH_3$-formation reaction is higher than that of Ni and Rh catalysts. Reforming gas containing $NH_3$ is known to poison Pt catalysts in Polymer Electrolyte Membrane Fuel Cells (PEMFCs) and also poison catalysts in preferential oxidation (PROX). In this study, Langmuir–Hinshelwood-based models of the $NH_3$-formation reaction considering $H_2$ and CO were proposed and compared with a simplified form of the Temkin–Pyzhev model for $NH_3$-formation rate. The kinetic parameters of each model were optimized by performing multi-objective function optimization on the experimental results using a tube-type reactor and the numerical results of a plug-flow one-dimension simple SR (steam-reforming) reactor.

**Keywords:** hydrogen; reformation; water–gas shift reaction; kinetic model; $NH_3$





## 1. Introduction

Fuel-cell systems are eco-friendly power-generation systems that convert chemical energy directly into electrical energy through an electrochemical reaction, and have the potential to achieve high operating efficiencies, to produce fewer pollutants, to have flexibility in fuel sources, and to make less noise [1–3]. However, hydrogen as a fuel does not exist in nature as a pure substance, so it must be manufactured from locally available resources [4].

In general, hydrogen-production methods include reforming technologies to obtain hydrogen from fossil fuels, gasification, thermochemical methods, and water electrolysis. Hydrogen production from fossil fuels such as natural gas and petroleum-based fuels, which have long been widely used in petroleum refining and petrochemical processes, is more economical than other production methods [5]. The process of generating hydrogen through fossil-fuel reforming involves catalytic decomposition of fossil fuels in the presence of water vapor or oxygen. The reforming methods include steam reforming, partial oxidation, and auto-thermal reforming [5].

The steam-reforming method is an endothermic reaction that causes hydrocarbons to react with steam and requires a heating source in the reactor. It has the advantage of achieving high efficiency by properly controlling the heat balance in the reactor and increasing the hydrogen concentration in the product gas [5].

However, if the feed gas supplied to the reformer contains nitrogen, an ammonia-production reaction occurs as a side reaction while the methane steam-reforming (SR) reaction takes place. In fuel-cell systems, ammonia damages not only the preferential oxidation (Prox) catalyst but also the membrane of the proton-exchange membrane fuel cell (PEMFC) [6]. A study by Fumihiro Watanbe et al. showed that especially Ru catalysts among precious metal catalysts produced a significant amount of ammonia [7]. Since

$Ru/Al_2O_3$ is one of the most used catalysts for steam reforming, it is necessary to predict the amount of ammonia produced for PEMFC using reformers with $Ru/Al_2O_3$ (RUA) catalysts to prevent the damage of the Prox but also the damage of the PEMFC from ammonia. In order to predict the ammonia-formation rate that occurs simultaneously with the steam-reforming reaction in a reformer on the surface of the RUA catalyst, it is necessary to conduct $NH_3$-formation experiments in a steam-reforming reactor with various concentrations of $N_2$ containing processed natural gas as a fuel. In this research, the kinetic model based on the Langmuir–Hinshelwood mechanism by Jon Geest Jakobsen is employed for the calculation of the concentration of each gas in the steam-reforming reaction [8–10] with optimized kinetic parameters for the steam-reforming reaction on RUA catalysts. To optimize the kinetic parameters, experiments were conducted for methane steam-reforming reactions in an RUA-catalyzed tube-type reactor and the reactor surface temperature, and the compositions of the reactor outlet gas were measured via a gas analyzer (NOVA prime-MRU, Germany). To optimize kinetic parameters with experimental data, a simplified numerical model of the tube-type plug-flow packed bed reactor was implemented. For each experimental condition (reactor-feed-gas composition, reactor surface temperature, charged-catalyst amount, etc.), the outlet composition was calculated through the numerical model [11]. During the calculation of the concentration, the kinetic parameters for the model were estimated using multi-objective function optimization. For the multi-objective function optimization, MATLAB's 'fgoalattain' function was implemented, which uses the sequential-quadratic-programming (SQP) method [12]. The objective function is the result of the numerical model that calculates the exit-gas composition according to each condition. The target value is the exit composition measured in the experiment.

Various kinetic models of the ammonia-formation reaction over metal catalysts have been compared such as the Temkin–Pyzhev equation [13–17], the power law, and the Langmuir–Hinshelwood-based models. The proposed Langmuir–Hinshelwood-mechanism-based model is derived by assuming that the vacant active site in the ammonia-production reaction is occupied by carbon monoxide and/or hydrogen produced in the steam-reforming reaction [8–10]. In a new kinetic model for ammonia formation in a steam-reforming reactor, we estimated the parameters of the ammonia kinetics model with the addition of adsorption terms of carbon monoxide and/or hydrogen with experimentally obtained data.

## 2. Results

### 2.1. Numerical Modeling

2.1.1. Steam-Reforming-Reactor Model

The SR-reactor-simulation model consists of a one-dimensional model (1D model) of a tube-type reactor assuming that there is no temperature gradient in the radial direction of the reactor and the flow of the feed gas is assumed to be a plug-flow reactor. The concept of the SR-reactor-simulation model is shown in Figure 1. The numerical model of a 1D plug-flow reactor is numerically analyzed using the mass-balance, the energy-balance, and reaction-rate equations [11,18]. For the mass balance, we used the input flow rate of reactants and the reactor-outlet-gas composition measured in the experiment. The energy balance was simplified by using experimentally measured reactor surface temperature assuming that there was no temperature difference in the coaxial direction of the reactor. Therefore, the reactor numerical model in this study performs numerical analysis of the reaction kinetics of each reaction model according to the input-gas flow rate, outlet-gas composition, and reaction surface temperature at given experimental conditions.

Two major reactions in the steam-reforming (SR) reactor are considered in the SR-simulation model for a steam-reforming (SR) reaction; they are shown as Equation (1) for the steam-reforming reaction and as Equation (2) for the water–gas-shift (WGS) reaction.

$$CH_4 + H_2O \leftrightarrow CO + 3H_2 \tag{1}$$

$$CO + H_2O \leftrightarrow CO_2 + H_2 \tag{2}$$

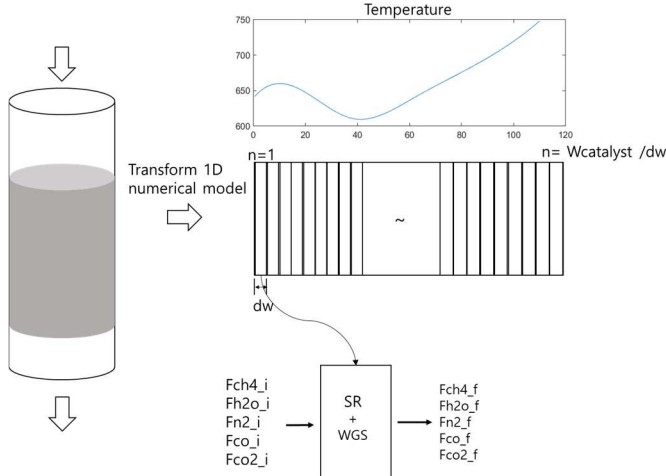

**Figure 1.** SR-reactor-simulation model.

To calculate the gas concentration at each location of the reactor, the one-dimensional reactor was virtually divided into N micro-cells charged with catalysts (Figure 1).

The outlet-gas composition of each micro-cell was calculated by combining the steam-reforming-reaction rate and the water–gas-shift (WGS)-reaction rate with the inlet-gas composition of the cell. The component mass-balance equations are listed as Equations (3)–(7).

$$F_{CH_4 out,n} = F_{CH_4 in,n} - R_{SR}(T_n, P_i) \times dw \tag{3}$$

$$F_{H_2O out,n} = F_{H_2O in,n} - R_{SR}(T_n, P_i) \times dw - R_{WGS}(T_n, P_i) \times dw \tag{4}$$

$$F_{H_2 out,n} = F_{H_2 in,n} + 3 \times R_{SR}(T_n, P_i) \times dw + R_{WGS}(T_n, P_i) \times dw \tag{5}$$

$$F_{CO out,n} = F_{CO in,n} + R_{SR}(T_n, P_i) \times dw - R_{WGS}(T_n, P_i) \times dw \tag{6}$$

$$F_{CO_2 out,n} = F_{CO_2 in,n} + R_{WGS}(T_n, P_i) \times dw \tag{7}$$

$F_{iout,n}$ is the outlet molar flow rate of chemical species i in cell n and $F_{i_{in,n}}$ is the inlet molar flow rate of chemical species i in cell n. $R_{SR}$ is the steam-reforming-reaction rate, $R_{WGS}$ is the water–gas-shift-reaction rate, and dw is the amount of catalyst in each cell. $T_n$ is the temperature in cell n and $P_i$ is the partial pressure of chemical species i.

The catalytic reaction rate of the steam-reforming reaction is given as Equations (8) and (9) by Jon Geest Jakobsen [8,9].

$$R_{SR} = \frac{A_1 \cdot exp(\frac{-E_1}{RT}) \cdot P_{CH_4} \cdot (1 - \beta_{SR})}{(1 + A_{CO} \cdot exp(\frac{-\Delta H_{CO}}{RT}) \cdot P_{CO} + A_H \cdot exp(\frac{-\Delta H_H}{RT}) \cdot P_{H_2}^{\frac{1}{2}})^2} \tag{8}$$

$$\beta_{SR} = \frac{P_{CO} P_{H_2}^3}{P_{CH_4} P_{H_2O}} \cdot \frac{1}{K_{P,SR}} \tag{9}$$

$$K_{P,SR} = SR_{equilibrium}(FCH_{4in,n}, FH_2O_{in,n}, FH_{2in,n}, FCO_{in,n}, FCO_{2in,n}) \tag{10}$$

$A_1$ is the Arrhenius constant and $E_1$ is the activation energy. $A_{co}$ and $A_H$ are the prefactors of the CO and H equilibrium constants, and $\Delta H_{CO}$ and $\Delta H_H$ are the enthalpies of adsorption. $\beta_{SR}$ is the approach to the SR-reaction equilibrium [8] and $P_i$ is the partial pressure of chemical species i. $K_{P,SR}$ is the thermodynamic equilibrium constant of the SR reaction.

The water–gas-shift reaction was calculated using the WGS kinetics model by Jian Sun as shown in Equations (11)–(13) [19].

$$R_{WGS} = \frac{A \cdot \exp\left(\frac{-E}{RT}\right) \cdot P_{CO}P_{H_2O} \cdot (1 - \beta_{WGS})}{\left(1 + A_{CO} \cdot \exp\left(\frac{-\Delta H_{CO}}{RT}\right) \cdot P_{CO}\right) \cdot \left(1 + A_{H_2} \cdot \exp\left(\frac{-\Delta H_{H2}}{RT}\right) \cdot P_{H_2}\right)} \tag{11}$$

$$\beta_{WGS} = \frac{P_{CO_2}P_{H_2}}{P_{CO}P_{H_2O}} \cdot \frac{1}{K_{P,WGS}} \tag{12}$$

$$K_{P,WGS} = WGS_{equilibrium}\left(FCH_{4in,n}, FH2O_{in,n}, FH2_{in,n}, FCO_{in,n}, FCO2_{in,n}\right) \tag{13}$$

A is the Arrhenius constant and E is the activation energy in the WGS reaction. $A_{co}$ and $A_{H2}$ are the prefactor of the CO and H equilibrium constants, and $\Delta H_{CO}$ and $\Delta H_{H2}$ are the enthalpies of adsorption. $B_{WGS}$ is the approach to the WGS-reaction equilibrium and $P_i$ is the partial pressure of chemical species i. $K_{P,WGS}$ is the thermodynamic equilibrium constant of the WGS reaction.

The initial values of the kinetics parameter of the reactor numerical model for the SR reaction and WGS reaction in experimental setup 1 (Section 3) are summarized in Table 1 [8,19].

**Table 1.** Initial values of the kinetic model for SR-reactor simulation [8,19].

| Steam-Reforming Kinetics Parameters | | | | | |
|---|---|---|---|---|---|
| $A_1$ (mol/g·h·bar) | $E_1$ (kJ/mol) | $A_{co}$ (bar$^{-1}$) | $\Delta H_{co}$ (kJ/mol) | $A_H$ (bar$^{-1/2}$) | $\Delta H_H$ (kJ/mol) |
| $4.39 \times 10^7$ | 107.9 | $2.19 \times 10^{-5}$ | $-87.4$ | $7.31 \times 10^{-6}$ | $-71$ |
| Water–Gas-Shift Kinetics Parameters | | | | | |
| A (mol/m$^3$.atm$^2$.s) | E (kJ/mol) | $A_{co}$ (atm$^{-1}$) | $\Delta H_{co}$ (kJ/mol) | $A_{H2}$ (atm$^{-1}$) | $\Delta H_{H2}$ (kJ/mol) |
| $2.00 \times 10^7$ | 43 | $9.40 \times 10^{-11}$ | $-100$ | $1.10 \times 10^{-10}$ | $-90$ |

2.1.2. Ammonia-Formation Kinetic Model in the SR Reactor

Two models were applied and compared for the prediction of the ammonia-formation rate in the SR reactor where a small amount of nitrogen is supplied to the SR reactor with methane. The most commonly used rate equation for ammonia synthesis is the Temkin–Pyzhev equation, proposed in 1940, which is derived by assuming that dissociative adsorption of nitrogen determines the rate and catalyst surface occupancy by nitrogen atoms is high [17]. Ozaki et al. proposed a rate equation (Equation (14)) that extends the original Temkin–Pyzhev equation when the surface occupancy of atomic nitrogen is not high [14,16]. In this study, the Temkin–Pyzhev equation extended by Ozaki (Equation (14)) was simplified to a power-law model as shown in Equations (15)–(18), assuming that there is an excess of hydrogen relative to ammonia in the reactor due to the high conversion of methane at the steam-reforming reaction [13,14,16,17]. Then, the calculation result was compared with a newly derived model.

$$R_{NH_3} = \frac{k'_A P_{N_2} - k'_B \left(P_{NH_3}\right)^2 / \left(P_{H_2}\right)^3}{\left[1 + k'_B \left(P_{NH_3}\right) / \left(P_{H_2}\right)^{3/2}\right]^{2\alpha}} \tag{14}$$

$$\text{If } P_{H_2} \gg P_{NH_3} \rightarrow k'_B \left(P_{NH_3}\right)^2 / \left(P_{H_2}\right)^3 \approx 0, \left[1 + k'_B \left(P_{NH_3}\right) / \left(P_{H_2}\right)^{\frac{3}{2}}\right]^{2\alpha} \approx 1 \tag{15}$$

$$R_{NH_3} = \frac{k'_A P_{N_2} - k'_B (P_{NH_3})^2 / (P_{H_2})^3}{\left[1 + k'_B (P_{NH_3}) / (P_{H_2})^{3/2}\right]^{2\alpha}} \cong k_{NH_3} \cdot P_{N_2} \tag{16}$$

$$R_{NH_3} = k_{NH_3} \cdot P_{N_2} \tag{17}$$

$$k_{NH_3} = A_{n2} \cdot \exp\left(\frac{-E_{n2}}{RT}\right) \tag{18}$$

Since the dominant reaction affecting the concentration of each component is the steam-reforming reaction and the WGS reaction, although these reactions proceed simultaneously with the ammonia-formation reaction inside the methane-steam-reforming reactor, the concentration of each component including hydrogen in a micro-cell in Figure 1 is calculated using Equations (8) and (11). The kinetics of the steam-reforming reaction over $Ru/ZrO_2$ catalysts by J.G. Jakobsen et al. [8] showed that CO and H atoms partially cover the catalyst surface at low temperatures, reducing the methane-steam-reforming activity [8,9]. Thus, the assumption that hydrogen and CO occupy the active sites on the catalyst with dissociative adsorption of nitrogen as the rate-determining step is applied for a new Langmuir–Hinshelwood-type kinetic model for ammonia formation in the SR reactor. The new kinetic model is explained in Equations (19)–(24) [8–10].

$$R_{NH_3} = k_{NH_3} P_{N_2} \theta_v^2 \tag{19}$$

$$\theta_v = (1 - \theta_N - \theta_{CO} - \theta_H) \tag{20}$$

$$K_{CO} = A_{CO} \cdot \exp\left(\frac{-\Delta H_{CO}}{RT}\right) \tag{21}$$

$$K_H = A_H \cdot \exp\left(\frac{-\Delta H_H}{RT}\right) \tag{22}$$

$$R_{NH_3} = \frac{k_{NH_3} P_{N_2} \cdot (1 - \beta_{NH_3})}{\left(1 + K_{CO} P_{CO} + K_H P_{H_2}^{1/2}\right)^2} \tag{23}$$

$$R_{NH_3} = \frac{k_{NH_3} P_{N_2} \cdot (1 - \beta_{NH_3})}{(1 + K_{CO} P_{CO})^2} \tag{24}$$

$$R_{NH_3} = \frac{k_{NH_3} P_{N_2} \cdot (1 - \beta_{NH_3})}{\left(1 + K_H P_{H_2}^{1/2}\right)^2} \tag{25}$$

$A_{n2}$ is the Arrhenius constant and $E_{n2}$ is the activation energy in ammonia-formation reactions. $A_{co}$ and $A_H$ are the prefactor of the CO and H equilibrium constants, and $\Delta H_{CO}$ and $\Delta H_H$ are the enthalpies of adsorption. $\beta_{NH3}$ is the approach to the ammonia-formation-reaction equilibrium, and $P_i$ is the partial pressure of chemical species i. $K_{P,WGS}$ is the thermodynamic equilibrium constant of the WGS reaction. $\theta_i$ is the active site on the catalyst surface of chemical species i.

The initial parameters of the power-law model for ammonia kinetics (Equation (16)) during parameter optimization using preliminary experimental data obtained in this study are summarized in Table 2.

**Table 2.** Initial parameters of the power-law model for $NH_3$ formation [20].

| NH₃ Kinetics Parameters | |
|---|---|
| $A_1$ (mol/g·h·bar) | $E_1$ (kJ/mol) |
| 5 | 80 |

### 2.1.3. Numerical Models for Estimating Kinetic Parameters

The process block diagram for optimizing the kinetic parameters of the SR reaction, the kinetic parameters of the WGS reaction, and of the kinetic parameters of the $NH_3$-formation reaction using the measured reactor temperatures and outlet-gas compositions in experiment set 1 (Section 3) and in experiment set 2 (Section 3) is shown in Figure 2.

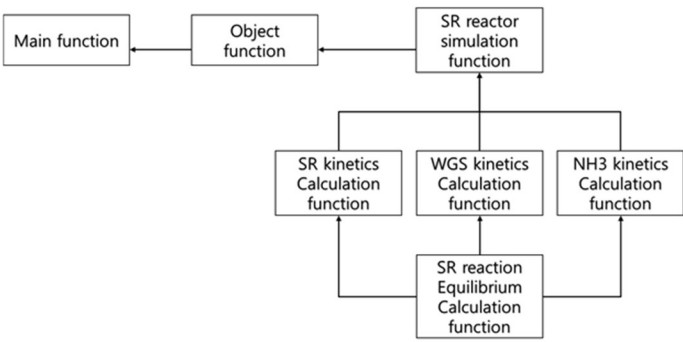

**Figure 2.** Process diagram for kinetic-parameter optimization.

The main function is to optimize the kinetics parameters of each reaction using MATLAB's multi-objective optimization function 'fgoalattain', which starts from the initial value of the kinetics parameters of the chemical reaction and optimizes the kinetic parameters to reduce the difference between the experimentally measured reactor-outlet-gas compositions at each experimental condition. The resulting value (objective value) is the calculated concentration of outlet flow, which is calculated via the objective function with adjusted kinetic parameters for the optimization process. The main function is made up of Equations (26)–(31).

Optimization variable

$$X = \left[SR_{parameters}, WGS_{parameters}, NH_{3.parameters}\right] \tag{26}$$

Kinetic parameters

$$SR_{parameters} = [A_1\ E_1\ A_{CO}\ \Delta H_{CO}\ A_H\ \Delta H_H] \tag{27}$$

$$WGS_{parameters} = \left[A\ E\ A_{CO}\ \Delta H_{CO}\ A_{H_2}\ \Delta H_{H_2}\right] \tag{28}$$

$$NH_{3.parameters} = [A_{n2}\ E_{n2}] \tag{29}$$

The kinetic parameters of each reaction are those of the SR-reaction kinetic Equation (8), the WGS-reaction kinetic Equation (11), and the $NH_3$-reaction kinetic Equation (18).

The constant is each experiment's conditions.

$$C_j = \left[f_{input}, T_{surface}, x_{catalyst}\right] \tag{30}$$

j is the experiment number and $C_j$ is the conditions for each experiment. $f_{input}$ is input flow rate, $T_{surface}$ is the temperature profile on the reactor surface, and $x_{catalyst}$ is the catalyst weight distribution based on location in the reactor.

Multi-objective optimization function

$$X_{optimal} = \text{fgoalattain}\left(F_{object}, X_{initial}, goal, weight.etc\right) \tag{31}$$

$X_{optimal}$ is the result of the optimized variable. $X_{initial}$ is the initial value of the optimization variable. Goal is the target value (experimental data). $F_{object}$ is the objective function. The objective function is generated as shown in Equation (32) in the objective-function block using the results of Equation (33), the SR-reactor numerical model.

The objective function runs the SR-reactor-simulation function with the kinetic parameters from the main function as variables and the reactant input flow rate and reactor surface temperature as constants for each experimental condition. The reactor-outlet-composition value, which is the calculation result of the SR-reactor-simulation function for each experimental condition, is returned to the objective function as a matrix for comparison with the objective value of the main function.

Objective function

$$F_{objet} = \begin{bmatrix} F_{SR}(X, C_1)_1 \\ \vdots \\ F_{SR}(X, C_n)_n \end{bmatrix} \tag{32}$$

SR-reactor numerical function

$$\left[F_{CH_4} \ F_{H_2O} \ F_{H_2} \ F_{CO} \ F_{CO_2} \ F_{N_2} \ F_{NH_3}\right] = F_{SR}\left(X, C_j\right)_j \tag{33}$$

$F_{SR}()_j$ is the SR-reactor-numerical-model function at the jth experimental condition.

The SR-reactor-simulation function consists of the SR-kinetics function, the WGS-kinetics function, the $NH_3$-kinetics function, and the equilibrium-calculation function that uses Gibb's minimization to calculate the equilibrium concentration for a given gas concentration and temperature. This function divides the total amount of catalyst in the reactor by the amount of micro-catalyst, dw, to make n calculation cells, as shown in the conceptual diagram in Figure 1. Each cell calculates the degree of reaction of each reactant and passes it to the next cell by calling the function named kinetics of each reaction with the temperature of the cell as an input variable in addition to the flow rate and concentration of the input gas from the previous cell. The calculation of each cell is performed sequentially, and the calculated concentration of the last nth cell is the outlet concentration of the reactor. The SR-reaction kinetics and the WGS-reaction kinetics for the SR-reactor simulation used the kinetics parameters estimated with the experimental data in experiment set 1. The kinetics parameters of the ammonia-production reaction were estimated via multi-objective function optimization in MATLAB using the SR-reactor numerical model as the objective function, with the measured gas-composition values at the reactor outlet in experiment set 2 as the objective values.

## 3. Experimental Results

### 3.1. Steam-Reforming and Water-Gas-Shift Reactions

3.1.1. Experimental Results

Experiment set 1 was performed to optimize the kinetic parameters of the Ru-catalyzed steam-reforming (SR) reaction and water–gas-shift (WGS) reaction. In the experiments, the surface temperature of the tube-type reactor was measured after the reaction reached a steady state as shown in Table 3.

The dry base composition after removing water from the reactor outlet gas for each experimental condition is summarized in Table 4.

**Table 3.** Reactor-surface-temperature measurements from experiment set 1.

| | Reactor-Surface-Temperature Measurements | | | | | |
|---|---|---|---|---|---|---|
| Experiment No. | TC1 | TC2 | TC3 | TC4 | TC5 | TC6 |
| 1 | 416.103 | 427.890 | 490.712 | 541.452 | 614.178 | 688.963 |
| 2 | 421.888 | 428.360 | 488.376 | 537.954 | 610.327 | 686.980 |
| 3 | 402.377 | 439.223 | 508.940 | 561.301 | 637.521 | 715.646 |
| 4 | 403.623 | 436.523 | 504.549 | 554.959 | 627.075 | 705.193 |
| 5 | 427.960 | 469.278 | 535.604 | 591.280 | 675.579 | 761.531 |
| 6 | 432.481 | 453.047 | 530.301 | 586.076 | 665.849 | 757.463 |
| 7 | 444.038 | 487.878 | 560.254 | 621.675 | 723.599 | 824.274 |
| 8 | 453.183 | 470.034 | 555.041 | 613.283 | 708.914 | 819.654 |

**Table 4.** SR-reactor-outlet-composition measurements from experiment set 1.

| | | Measured Composition of the Reactor Outlet Gas (Dry Base) | | | | | |
|---|---|---|---|---|---|---|---|
| Experiment No. | Furnace Setup Temp. | $F_{CH4}$ (mol/) | $CH_4$ (%) | $H_2$ (%) | CO (%) | $CO_2$ (%) | $CH_4$ Conversion (%) |
| 1 | 500 | 2.6771 | 6.4341 | 71.7208 | 11.9954 | 11.0434 | 78.18 |
| 2 | 500 | 2.9449 | 6.8156 | 71.5951 | 11.1548 | 11.5788 | 76.92 |
| 3 | 520 | 2.6771 | 4.2164 | 73.0609 | 14.2023 | 9.8340 | 85.06 |
| 4 | 520 | 3.4803 | 5.2540 | 72.6521 | 11.8964 | 11.3181 | 81.56 |
| 5 | 550 | 2.6771 | 2.0439 | 74.2690 | 17.4827 | 7.8603 | 92.55 |
| 6 | 550 | 3.4803 | 2.1483 | 74.6066 | 15.0817 | 9.6202 | 91.99 |
| 7 | 580 | 2.6771 | 0.3226 | 75.3647 | 19.4244 | 6.8561 | 98.80 |
| 8 | 580 | 3.7480 | 0.4790 | 75.6000 | 17.4718 | 8.2331 | 98.17 |

### 3.1.2. Calculation Results

A 1D plug-flow reactor model was applied to optimize the kinetic parameters of Equation (8) for the SR-reaction rate and the kinetic parameters of Equation (11) for the WGS-reaction rate. To optimize the kinetic parameters, the calculated composition of the outlet gas and the measured composition of the outlet gas were compared to minimize the difference. The optimized kinetic parameters are summarized in Table 5. A comparison between the calculated concentration of each gas in the outlet gas and the measured concentration of it in experiment set 1 is summarized in Figure 3.

**Table 5.** Estimated SR-reaction and WGS-reaction kinetic parameters.

| Steam-Reforming Kinetics Parameters | | | | | |
|---|---|---|---|---|---|
| $A_1$ (mol/g.h.bar) | $E_1$ (kJ/mol) | $A_{co}$ (bar$^{-1}$) | $\Delta H_{co}$ (kJ/mol) | $A_H$ (bar$^{-1/2}$) | $\Delta H_H$ (kJ/mol) |
| $4.3785 \times 10^7$ | $1.2768 \times 10^2$ | $9.0900 \times 10^{-5}$ | $-9.5343 \times 10^1$ | $6.7500 \times 10^{-6}$ | $-7.6548 \times 10^1$ |
| **WGS Kinetics Parameters** | | | | | |
| A (mol/m$^3$.atm$^2$.s) | E (kJ/mol) | $A_{co}$ (atm$^{-1}$) | $\Delta H_{co}$ (kJ/mol) | $A_{H2}$ (atm$^{-1}$) | $\Delta H_{H2}$ (kJ/mol) |
| $1.9991 \times 10^7$ | $4.2383 \times 10^1$ | $3.6400 \times 10^{-11}$ | $-9.9403 \times 10^1$ | $3.6000 \times 10^{-11}$ | $-8.8443 \times 10^1$ |

The graph in Figure 3 displays the measured values from experiment set 1 on the *x*-axis and the calculated values from the numerical model on the *y*-axis. As shown in Figure 3, the kinetic model with optimized parameters can predict most gas concentrations within a 20% error, except methane, hydrogen, and $CO_2$. For hydrogen and $CO_2$, the model can predict each concentration within a 5% error. In the case of methane-concentration prediction, the prediction error is less than 10% in the low-conversion region. As methane conversion increased from low to high levels, methane concentrations were under-predicted, resulting in an average error of 42.64% (Table 6).

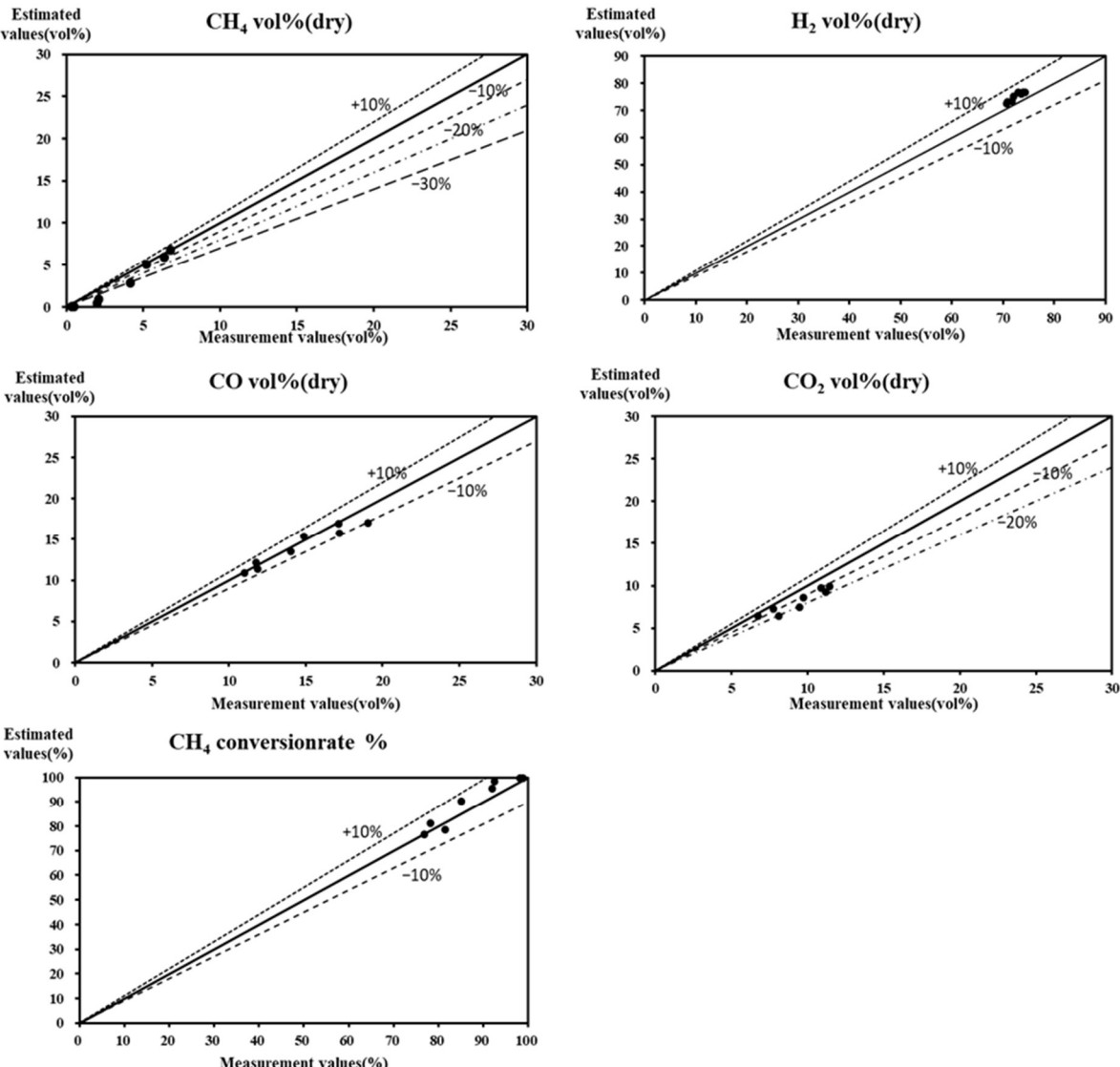

**Figure 3.** SR- and WGS-reaction-simulation results vs. experiment results in Table 4. Black circle (●) indicate measured and calculated values under the same experimental conditions.

**Table 6.** SR- and WGS-reaction-simulation results vs. experiment results average errors.

| Average Error (%) | | | | |
|---|---|---|---|---|
| $CH_4$ | $H_2$ | CO | $CO_2$ | $CH_4$ Conversion Rate |
| 42.64 | 3.32 | 4.46 | 13.16 | 3.31 |

*3.2. NH₃ Formation Reaction*

3.2.1. Experimental Results

The temperature-measurement scheme is shown in Section 4 Experiments, and the measured temperature values are shown in Table 7. TC1 is the topmost part of the reactor surface of the catalyst bed where the gas flows in, and TC3 measures the surface temperature of the bottommost part of the catalyst bed in the reactor.

**Table 7.** Reactor-surface-temperature-measurement results from experiment set 2.

| Reactor-Surface-Temperature Measurements | | | |
|---|---|---|---|
| Experiment No. | TC1 | TC2 | TC3 |
| 1 | 434.2252 | 553.9525 | 595.2163 |
| 2 | 458.5575 | 586.2754 | 632.4215 |
| 3 | 483.6517 | 619.8598 | 672.3031 |
| 4 | 499.0160 | 644.0106 | 700.7845 |
| 5 | 513.5956 | 668.8458 | 731.3515 |
| 6 | 530.3918 | 710.7519 | 789.1044 |
| 7 | 548.0571 | 755.8316 | 834.4375 |
| 8 | 440.0291 | 556.2471 | 605.9996 |
| 9 | 463.8518 | 588.7224 | 646.0989 |
| 10 | 487.6856 | 622.2952 | 686.5030 |
| 11 | 501.8929 | 645.3193 | 713.8137 |
| 12 | 516.5297 | 670.3384 | 747.8994 |
| 13 | 537.5016 | 712.1737 | 798.3009 |
| 14 | 548.4273 | 744.9738 | 830.3618 |
| 15 | 437.2601 | 560.6386 | 610.6106 |
| 16 | 462.3212 | 593.1973 | 644.3216 |
| 17 | 486.0982 | 629.5613 | 679.6483 |
| 18 | 504.6832 | 652.3975 | 713.9694 |
| 19 | 518.6271 | 675.0642 | 736.6576 |
| 20 | 541.1640 | 714.3720 | 786.7085 |
| 21 | 553.4275 | 741.6453 | 815.8044 |
| 22 | 443.5432 | 560.9095 | 602.7401 |
| 23 | 470.3663 | 593.2220 | 644.9947 |
| 24 | 496.1722 | 625.3554 | 687.1435 |
| 25 | 513.8378 | 641.2851 | 711.8383 |
| 26 | 528.0075 | 663.2032 | 746.5344 |
| 27 | 540.0961 | 702.7927 | 810.3330 |

The composition of the reactor outlet gas in experiment set 2 was measured in a gas analyzer (NOVA prime, MRU) after the water vapor contained in the outlet gas was removed from the cooler. Ammonia concentration was determined by condensing the outlet gas in a cooler for 20 to 30 min and measuring the ammonia dissolved in the condensate using colorimetric determination. The total molar flow rate was calculated using the volumetric composition of the dry gas measured on the gas analyzer for the outlet gas while the condensate was collected over a period of time. The ammonia flow rate was calculated by adding the amount of ammonia in the condensate and exhaust gases. The results of measuring the outlet-gas concentrations of the SR reactor are listed in Table 8.

**Table 8.** Reactor-outlet-composition measurements from experiment set 2.

| Reactor-Outlet-Composition Measurement | | | | | | | |
|---|---|---|---|---|---|---|---|
| Experiment No. | Furnace Setup Temp. | $F_{CH4}$ (mol/h) | $CH_4$ (%) | $H_2$ (%) | CO (%) | $CO_2$ (%) | $NH_3$ (ppm) | $CH_4$ Conversion (%) |
| 1 | 520 | 2.8295 | 11.4679 | 69.6571 | 5.8750 | 12.5571 | 2.1949 | 61.65 |
| 2 | 550 | 2.7278 | 7.5450 | 72.8300 | 8.3900 | 11.3350 | 3.8545 | 72.32 |
| 3 | 580 | 2.6314 | 4.3583 | 75.4958 | 11.3250 | 9.8083 | 6.9101 | 82.90 |
| 4 | 600 | 2.5725 | 2.7750 | 76.5813 | 13.1533 | 8.6125 | 10.4133 | 88.67 |
| 5 | 620 | 2.5136 | 1.4444 | 77.4389 | 14.7889 | 7.4667 | 13.9303 | 93.92 |
| 6 | 650 | 2.4333 | 0.3067 | 78.1733 | 16.2533 | 6.5000 | 15.2267 | 98.66 |
| 7 | 680 | 2.3557 | 0.0765 | 78.3412 | 16.5706 | 6.2882 | 11.7188 | 99.65 |
| 8 | 520 | 2.8054 | 12.8150 | 68.4650 | 5.8350 | 12.4400 | 2.6023 | 58.78 |
| 9 | 550 | 2.7037 | 8.2000 | 72.0348 | 8.4957 | 11.1565 | 4.2844 | 70.57 |
| 10 | 580 | 2.6100 | 5.2609 | 74.0696 | 11.5348 | 9.2435 | 8.7123 | 79.79 |

**Table 8.** *Cont.*

| Reactor-Outlet-Composition Measurement | | | | | | | |
|---|---|---|---|---|---|---|---|
| Experiment No. | Furnace Setup Temp. | $F_{CH4}$ (mol/h) | $CH_4$ (%) | $H_2$ (%) | CO (%) | $CO_2$ (%) | $NH_3$ (ppm) | $CH_4$ Conversion (%) |
| 11 | 600 | 2.5484 | 3.0895 | 75.7000 | 13.2316 | 8.2421 | 13.2180 | 87.42 |
| 12 | 620 | 2.4922 | 1.2292 | 75.7125 | 14.6250 | 7.4583 | 17.9330 | 94.73 |
| 13 | 650 | 2.4119 | 0.2040 | 76.5120 | 15.6320 | 6.8480 | 19.3508 | 99.12 |
| 14 | 670 | 2.3610 | 0.0600 | 76.6500 | 15.7000 | 6.8100 | 17.4777 | 99.73 |
| 15 | 520 | 2.7840 | 11.9455 | 67.3864 | 5.7591 | 12.4364 | 3.3526 | 60.36 |
| 16 | 550 | 2.6823 | 7.3000 | 70.8850 | 8.3000 | 11.2950 | 5.8103 | 72.86 |
| 17 | 580 | 2.5859 | 4.7913 | 72.6261 | 11.5261 | 9.1565 | 11.9798 | 81.20 |
| 18 | 600 | 2.5297 | 2.6565 | 74.2217 | 13.2696 | 8.2130 | 20.2724 | 88.98 |
| 19 | 620 | 2.4708 | 1.2875 | 75.2292 | 14.7167 | 7.2375 | 25.2017 | 94.45 |
| 20 | 650 | 2.3905 | 0.2760 | 75.9200 | 15.8720 | 6.4920 | 26.0380 | 98.76 |
| 21 | 670 | 2.3396 | 0.1040 | 76.0000 | 16.2080 | 6.2480 | 20.4790 | 99.56 |
| 22 | 520 | 2.6769 | 12.4650 | 66.0600 | 5.4950 | 12.1950 | 4.2351 | 58.67 |
| 23 | 550 | 2.6769 | 7.5909 | 70.2682 | 8.0045 | 11.2500 | 8.4266 | 71.72 |
| 24 | 580 | 2.6769 | 4.2409 | 72.8091 | 10.6273 | 9.8136 | 13.4893 | 82.82 |
| 25 | 600 | 2.6769 | 2.1500 | 74.3727 | 12.1773 | 8.9455 | 19.1980 | 90.76 |
| 26 | 620 | 2.6769 | 0.9952 | 75.1143 | 13.5476 | 8.0381 | 25.0476 | 95.57 |
| 27 | 650 | 2.6769 | 0.3043 | 75.6130 | 15.3304 | 6.8522 | 21.5246 | 98.67 |

### 3.2.2. Comparison between Calculated Values and Measured Valued

The formation rate of $NH_3$ was calculated via two kinetic models and the calculation results were compared with experimental data. The first model is the power-law model, and the second model is the newly derived kinetic model based on the Langmuir–Hinshelwood mechanism. The power-law model is given as Equations (14)–(17) and the optimized kinetic parameters in Equation (17) are in Table 9. The concentration of $NH_3$ in the outlet gas was calculated via the power-law model and compared with experimental data in experiment set 2; this is summarized in Figure 4. As mentioned, the concentrations of the remaining components are calculated with Equations (8) and (11).

**Table 9.** $NH_3$ power-law parameters results.

| $NH_3$ Kinetics Parameters Results | |
|---|---|
| $A_{n2}$ (mol/g.h.bar) | $E_{n2}$ (kJ/mol) |
| 4.9809 | $8.0787 \times 10^1$ |

The $NH_3$ concentration of SR-reactor outlet gas was calculated via the power-law model and compared with experimental data obtained in experiment set 2. Then, the concentration of $NH_3$ was calculated via the newly derived Langmuir–Hinshelwood-mechanism-based kinetic model and compared with experimental data. For the calculation of the rest of the species, Equation (8) by Geest Jakobsen for the SR-reaction rate [8,10] and Equation (11) by Jian Sun for the WGS-reaction rate [19] were employed. The optimized kinetic parameters for the power-law model are listed in Table 9, the average error for power-law model are listed in Table 10 and those for the new kinetic model are in Tables 11–14.

**Table 10.** Average errors in Figure 4 results.

| Average Error (%) | | | | | |
|---|---|---|---|---|---|
| $CH_4$ | $H_2$ | CO | $CO_2$ | $NH_3$ | $CH_4$ Conversion Rate |
| 43.99% | 4.28% | 8.84% | 3.92% | 88.29% | 8.53% |

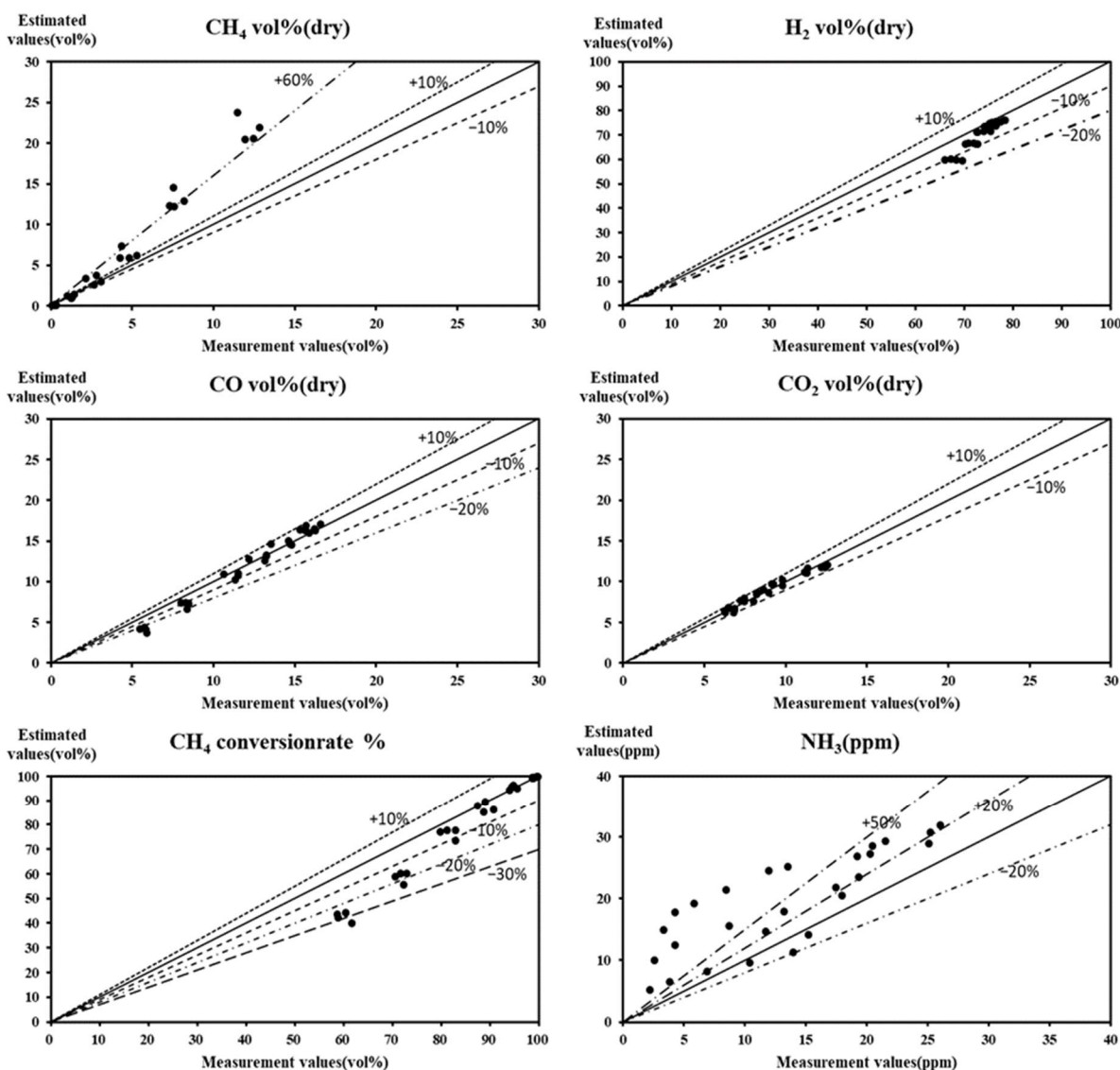

**Figure 4.** NH$_3$ power-law model reactor exit composition calculations vs. experimental results. Black circle (●) indicate measured and calculated values under the same experimental conditions.

**Table 11.** Optimized parameters for the Langmuir–Hinshelwood-based kinetic model for ammonia formation (adsorption term fixed).

| NH$_3$ Kinetics Parameters | | | | | |
|---|---|---|---|---|---|
| A$_{n2}$ (mol/g.h.bar) | E$_{n2}$ (kJ/mol) | A$_{co}$ (bar$^{-1}$) | $\Delta$H$_{co}$ (kJ/mol) | A$_H$ (bar$^{-1/2}$) | $\Delta$H$_H$ (kJ/mol) |
| 4.7953 | $8.2667 \times 10^1$ | $9.0900 \times 10^{-5}$ | $-9.5343 \times 10^1$ | $6.7500 \times 10^{-6}$ | $-7.6548 \times 10^1$ |

**Table 12.** Optimized parameters for the Langmuir–Hinshelwood-based model (CO-adsorption term fixed).

| NH$_3$ Kinetics Parameters | | | |
|---|---|---|---|
| A$_{n2}$ (mol/g.h.bar) | E$_{n2}$ (kJ/mol) | A$_{co}$ (bar$^{-1}$) | $\Delta$H$_{co}$ (kJ/mol) |
| 5.3597 | $8.7339 \times 10^1$ | $9.0900 \times 10^{-5}$ | $-9.5343 \times 10^1$ |

**Table 13.** Optimized parameters for the Langmuir–Hinshelwood-based model (H adsorption term fixed).

| NH$_3$ Kinetics Parameters | | | |
|---|---|---|---|
| A$_{n2}$ (mol/g.h.bar) | E$_{n2}$ (kJ/mol) | A$_H$ (bar$^{-1/2}$) | ΔH$_H$ (kJ/mol) |
| 4.9920 | $8.0381 \times 10^1$ | $6.7500 \times 10^{-6}$ | $-7.6548 \times 10^1$ |

**Table 14.** Average errors for the three Langmuir–Hinshelwood-based models.

| NH$_3$ Average Error | | | |
|---|---|---|---|
| Adsorption Term | CO and H | CO | H |
| Error (%) | 57.05% | 68.14% | 54.02% |

The ammonia production measured in experiment 2 and the ammonia production calculated from the 1D simple SR-reactor numerical model are plotted. The simplified power-law model is explained in Equations (14)–(16).

For the 1D simple SR-reactor numerical model, the ammonia-production-reaction rate was calculated using a power-law model that simplifies the Temkin–Pyzhev equation by assuming that the reactant gas in the SR reactor contains a relative excess of hydrogen compared to nitrogen. The optimized kinetics parameters of the power-law model of the ammonia-formation rate are shown in Table 9. Using the parameters in Table 9, the ammonia-production rate was applied to a simple SR-reactor simulation to calculate the outlet composition and ammonia production of the SR reactor. A graph comparing the values calculated from the simple SR-reactor simulation and the outlet composition measured in experiment set 2 is shown in Figure 4.

In the high conversion region, the experimental and calculated values are in good agreement within 10% error for the outlet-flow-concentration prediction and the methane-conversion rate. In the low-conversion region, the calculated value tends to be lower than the measured value, so the average error for methane concentration is 44%. In the case of carbon monoxide and carbon dioxide as shown in Figure 4, the difference between the experimental value and the measured value is within the error range of 10~20%. The ammonia-concentration-prediction results of the outlet flow show an error of about 20% at the higher-ammonia-concentration region (above 15 ppm), but at the lower-concentration regions (below 15 ppm) the scatter increases, and the prediction shows higher calculated concentrations. The average error of the predicted value of ammonia concentration is 88% (Table 10).

The results of optimizing all parameters of the Langmuir–Hinshelwood-based-model kinetics derived by applying the assumption that CO and H occupy the vacant active sites during the SR reaction to the ammonia-formation rate are shown in Table 11 (Equations (18) and (21)–(23)).

In the Langmuir–Hinshelwood-based model of ammonia formation, which considers the active site occupancy of CO and H, the parameters of the adsorption of CO and H in the denominator are fixed to the same values as the parameter values of the adsorption terms in the kinetic model of the SR reaction, since the reactant consumption of the ammonia reaction is very much smaller and the same catalyst is used for both cases.

A graph comparing the ammonia composition at the reactor outlet calculated by applying the Langmuir–Hinshelwood-based model in a 1D simple SR-reactor simulation using the parameters in Table 11 and the measured ammonia composition at the reactor outlet in experiment set 2 is shown in Figure 5.

A comparison between the calculated concentration of ammonia and the measured concentration is shown in Figure 5. The *x*-axis of the graph is the measured ammonia concentration in experiment set 2, and the *y*-axis is the concentration calculated via the Langmuir–Hinshelwood-based model with optimized parameters that considers the ad-

sorption of CO and H. The results in Figure 5 show that the model under-predicts the ammonia concentration.

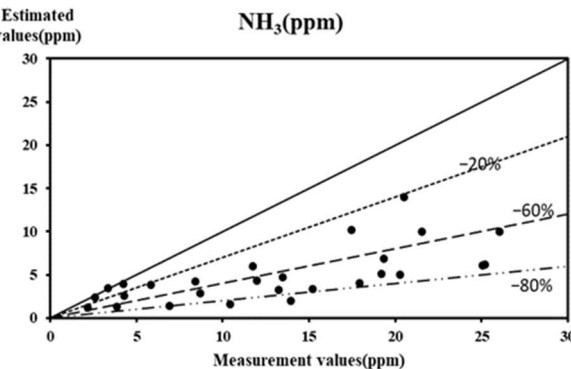

**Figure 5.** NH$_3$ calculation versus experiment result (The Langmuir–Hinshelwood-based model). Black circle (●) indicate measured and calculated values under the same experimental conditions.

The Langmuir–Hinshelwood-based model of the ammonia-formation reaction, derived by applying the assumption that carbon monoxide formed via the SR reaction has a large effect on occupying the vacant active site of the catalyst during the ammonia-formation reaction, was applied to the SR-reactor numerical model to estimate the ammonia-formation-rate parameters, and the results are shown in Table 12. The parameters of the carbon-monoxide-adsorption term in the denominator of this model are the values of the carbon-monoxide-adsorption term from the SR kinetics in Table 5.

A comparison between the calculated ammonia concentration at the outlet of the SR reactor and the measured concentration at the outlet flow in experiment set 2 is shown in Figure 6.

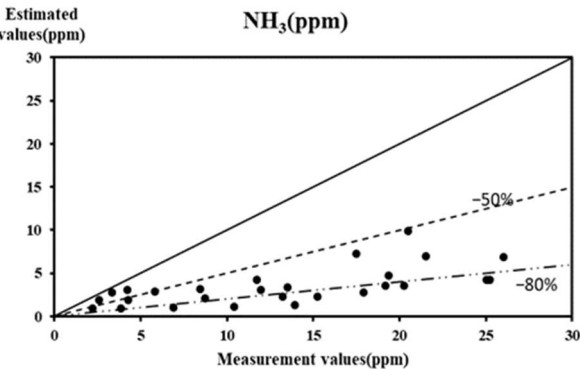

**Figure 6.** NH$_3$ calculation versus experiment result (CO-adsorption term only). Black circle (●) indicate measured and calculated values under the same experimental conditions.

The *x*-axis of the graph is the measured concentration of NH$_3$ of experiment set 2, and the *y*-axis is the concentration of the ammonia calculated by applying the parameter estimates of the Langmuir–Hinshelwood-based model that considers the adsorption of CO. The results in Figure 6 show that calculated concentrations are smaller than measured concentrations. Both the Langmuir–Hinshelwood-based model for the ammonia-formation reaction in the SR reactor (shown in Section 4 Experiments), assuming CO and H as adsorption terms, and the Langmuir–Hinshelwood-based model assuming only CO as adsorption term, show that the calculated ammonia concentrations in the flow are lower than the experimentally measured concentrations. Thus the parameters of the ammonia formation were estimated by applying the Langmuir–Hinshelwood model where only H is considered as the adsorption term. The Langmuir–Hinshelwood-based model of the ammonia-formation reaction, derived by applying the assumption that hydrogen has a large effect on occupying the vacant active sites of the catalyst during the ammonia-formation

reaction, is listed in the Equations (18), (22) and (25). The parameters of the hydrogen term in the denominator of this model are fixed using the values of the hydrogen term in the SR kinetics in Table 5, and the results of the optimized parameters of the ammonia-formation rate are summarized in Table 13.

The comparison between calculated ammonia production at the outlet of the SR reactor and measured concentrations in experiment set 2 is shown in Figure 7.

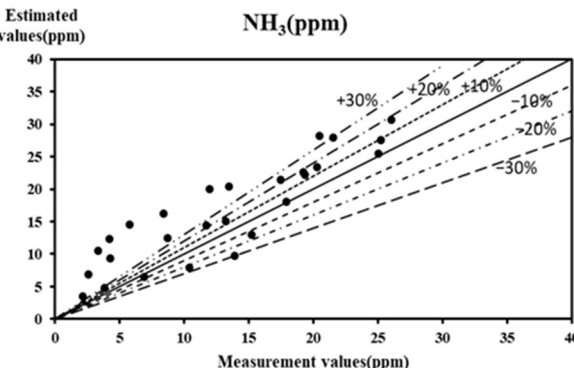

**Figure 7.** $NH_3$ calculation versus experiment (H adsorption term only). Black circle (●) indicate measured and calculated values under the same experimental conditions.

The *x*-axis of the graph is the measured ammonia concentrations in experiment set 2, and the *y*-axis is calculated ammonia-formation concentrations in a 1D simple SR-reactor numerical model via the Langmuir–Hinshelwood-based model, which takes into account the adsorption of H. The results in Figure 7 show that the slope of the trend line is close to 1, indicating that the trend of the ammonia-formation rate from the numerical model and those from the experiment are the same. Compared to the estimation results of the previous three models as shown in Table 14, the Langmuir–Hinshelwood-based model considering hydrogen adsorption predict the ammonia-formation rate the best since it follows the trend of experimental data.

## 4. Experiments

Two sets of RUA ($Ru/Al_2O_3$)-catalyzed methane steam-reforming reaction experiments were conducted in a tube-type reactor. Experimental set 1 was conducted to optimize the kinetic parameters of the steam-reforming (SR) reaction and water–gas-shift (WGS) reaction (side reaction) with Ru catalyst in an SR reactor, and experimental set 2 was conducted to measure the ammonia-formation rate in the steam-reforming reaction as a side reaction with $N_2$ containing process natural gas (PNG). Information about RUA ($Ru/Al_2O_3$) catalysts used for both sets of experiments is shown in Table 15. An electric furnace was installed in an SR reactor to supply the heat needed for the reaction. The reactor-outlet-gas composition was measured via a gas analyzer (NOVA prime-MRU model, Neckarsulm, Germany).

**Table 15.** Catalyst information.

| Catalyst | Size | Shape | Content | Support |
|----------|------|-------|---------|---------|
| Ru | 3 mm | Sphere | 2 (wt%) | $\alpha$-$Al_2O_3$ |

The schematic diagram of the experimental setup is in Figure 8. A mass flow controller (MFC) made by Linetech was used for the flow control of the feed gas and a metering pump was used for the supply of $H_2O$. The reactor was loaded with 109.56 g of Ru catalyst for SR reaction.

In experimental set 1, to estimate the kinetics parameters of SR and WGS reactions, the S/C ratio was kept at 2.5. The experimental conditions are summarized in Table 16.

Schematic diagram of experiment    Experiment procedure

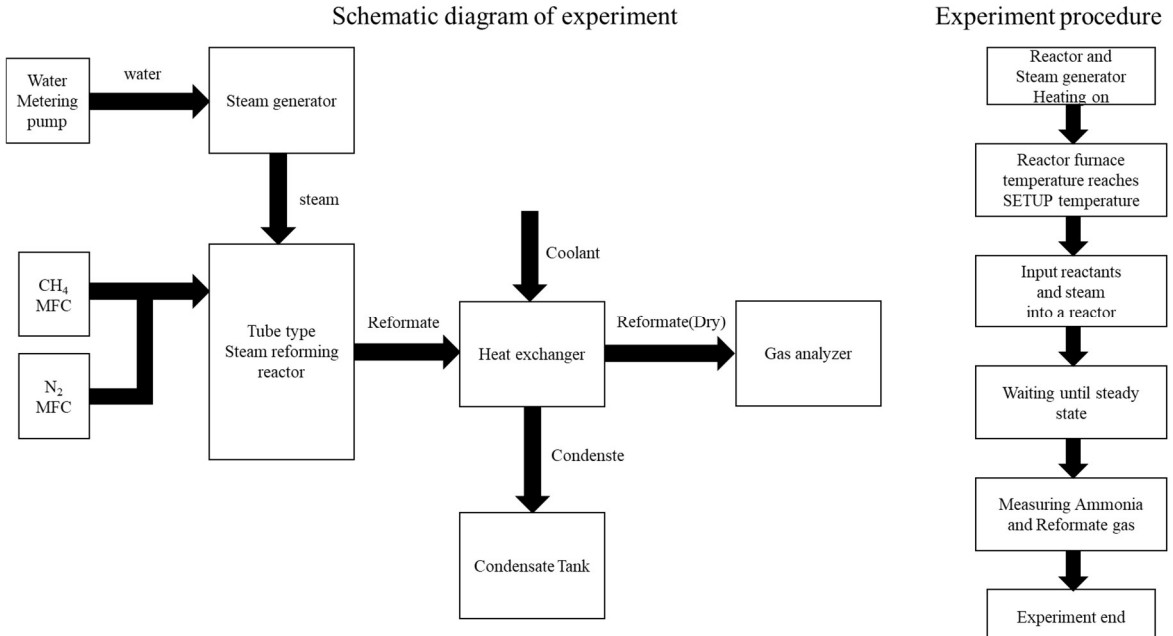

**Figure 8.** Schematic diagram of experiment and procedure.

**Table 16.** Experiment 1 conditions for SR and WGS kinetic parameters estimation.

| Experimental Conditions | | | |
|---|---|---|---|
| Experiment | Furnace Temperature (°C) | $F_{CH4\_i}$ (mol/h) | $F_{H2O\_i}$ (mol/h) |
| 1 | 500 | 2.6771 | 6.69 |
| 2 | 500 | 2.9449 | 7.36 |
| 3 | 520 | 2.6771 | 6.69 |
| 4 | 520 | 3.4803 | 8.70 |
| 5 | 550 | 2.6771 | 6.69 |
| 6 | 550 | 3.4803 | 8.70 |
| 7 | 580 | 2.6771 | 6.69 |
| 8 | 580 | 3.7480 | 9.37 |

The reactor surface temperatures were measured by attaching six k-type thermocouples to the reactor surface as shown in Figure 9.

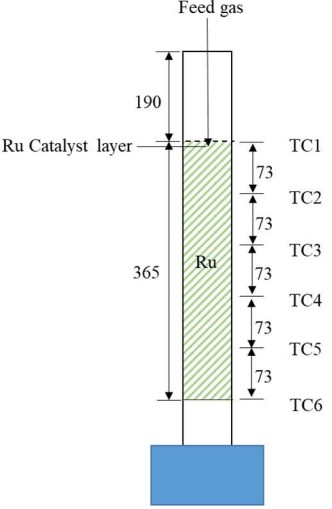

**Figure 9.** Reactor $T_C$ locations for SR and WGS kinetic experiments.

To measure the ammonia-formation rate in experiment set 2, the flow rate of $CH_4$ supplied to the reactor was 2.3~2.8 mol/h and mixed with $N_2$ at concentrations of 3, 6, 9, and 10% as shown in Table 17. During experiments, the S/C ratio was fixed at 2.5.

**Table 17.** Experimental conditions for $NH_3$-formation experiment set 2.

| Experimental Conditions for $NH_3$ Formation | | | | |
|---|---|---|---|---|
| Experiment No. | Furnace Temperature (°C) | $F_{CH4\_i}$ (mol/h) | $F_{H2O\_i}$ (mol/h) | $N_2/CH_4$ |
| 1 | 520 | 2.8295 | 7.0738 | 0.03 |
| 2 | 550 | 2.7278 | 6.8195 | 0.03 |
| 3 | 580 | 2.6314 | 6.5785 | 0.03 |
| 4 | 600 | 2.5725 | 6.4313 | 0.03 |
| 5 | 620 | 2.5136 | 6.2840 | 0.03 |
| 6 | 650 | 2.4333 | 6.0833 | 0.03 |
| 7 | 680 | 2.3557 | 5.8893 | 0.03 |
| 8 | 520 | 2.8054 | 7.0135 | 0.06 |
| 9 | 550 | 2.7037 | 6.7593 | 0.06 |
| 10 | 580 | 2.6100 | 6.5250 | 0.06 |
| 11 | 600 | 2.5484 | 6.3710 | 0.06 |
| 12 | 620 | 2.4922 | 6.2305 | 0.06 |
| 13 | 650 | 2.4119 | 6.0298 | 0.06 |
| 14 | 670 | 2.3610 | 5.9025 | 0.06 |
| 15 | 520 | 2.7840 | 6.9600 | 0.09 |
| 16 | 550 | 2.6823 | 6.7058 | 0.09 |
| 17 | 580 | 2.5859 | 6.4648 | 0.09 |
| 18 | 600 | 2.5297 | 6.3243 | 0.09 |
| 19 | 620 | 2.4708 | 6.1770 | 0.09 |
| 20 | 650 | 2.3905 | 5.9763 | 0.09 |
| 21 | 670 | 2.3396 | 5.8490 | 0.09 |
| 22 | 520 | 2.6769 | 6.6923 | 0.10 |
| 23 | 550 | 2.6769 | 6.6923 | 0.10 |
| 24 | 580 | 2.6769 | 6.6923 | 0.10 |
| 25 | 600 | 2.6769 | 6.6923 | 0.10 |
| 26 | 620 | 2.6769 | 6.6923 | 0.10 |
| 27 | 650 | 2.6769 | 6.6923 | 0.10 |

To measure the reactor surface temperatures, three k-type thermocouples were installed on the surface of the catalyst layer as shown in Figure 10.

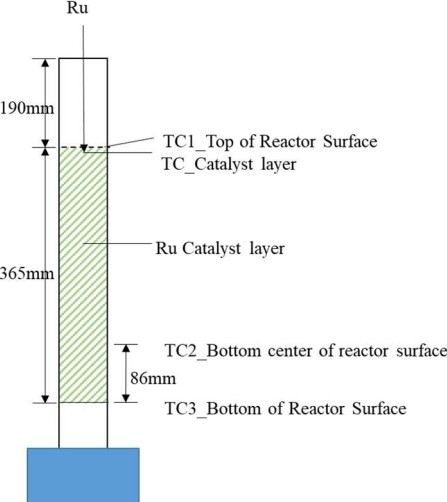

**Figure 10.** Installed positions of thermocouples in experiment set 2.

To measure the ammonia-formation rate, the outlet gas of the SR reactor was cooled and condensed, and condensate was collected. Additionally, the flow rate of heat-exchanger outlet gas was measured to calculate the ammonia composition of the outlet gas. The total amount of ammonia produced over time was measured by the summation of the amount of ammonia dissolved in a condensate over time and the amount of ammonia in the outlet gas over time. The amount of ammonia dissolved in the condensate was quantitatively measured via the colorimetric method using a Compact Ammonia Duo coloricmeter with Palintest. The composition of each component in the outlet gas was measured via a gas analyzer (NOVA prime-MRU). The molarity of the SR reactor outlet gas was calculated with the measured outlet-gas composition and the measured outlet gas flow rate. Using the exit gas molarity and the amount of ammonia in the condensate, the ammonia flow rate produced by the SR reactor was calculated.

### 5. Conclusions

Ru catalyst causes $N_2$ to react with $H_2$ to form ammonia at steam-reforming conditions and if supplied to the PEMFC, produced ammonia would seriously damage the PEMFC fuel cell. The European sort contains 1–5% or more $N_2$ in natural gas. Thus, to use a fuel-processing system for hydrogen production supplied to PEMFC in Europe, it is essential to investigate the ammonia concentration in a fuel-processing product gas to prevent ammonia formation. The $Ru/Al_2O_3$-catalyzed-SR-reaction and ammonia-formation-reaction experiments were conducted in a tube-type reactor and then numerically modeled by combining a simple 1D-reactor-simulation model with a kinetic model. The numerical modeling used a power-law model that simplifies the traditional ammonia-generation kinetics by Temkin–Pyzhev with the assumption that it reacts in an excess amount of hydrogen, and a Langmuir–Hinshelwood-based model with the assumptions that methane dissociative adsorption is the rate-limiting step and adsorbed CO and H cover the active sites of the catalyst affecting the overall activity. To optimize the kinetic parameters for each model, the multi-objective optimization method was used. The results of the numerical model showed that the model with the assumption that $H_2$ adsorbs and partially covers the active sites of the catalyst gave a better correlation with the experimental data. The new model would predict the ammonia concentration in the outlet flow of the fuel-processing system under various conditions and this information would be used to find operating conditions for the reformer to reduce the ammonia formation.

**Author Contributions:** Conceptualization, C.K. and S.L.; Formal analysis, C.K. and S.L.; Investigation, C.K. and J.L.; Writing—original draft, C.K.; Writing—review & editing, S.L.; Supervision, S.L.; Project administration, S.L.; Funding acquisition, S.L. All authors have read and agreed to the published version of the manuscript.

**Funding:** This research was funded by the Korea Institute of Energy Technology Evaluation and Planning (KETEP), grant number 20183010032400 and 20203040030110.

**Data Availability Statement:** New own data is contained within the article. Other data have been cited.

**Acknowledgments:** The authors are thankful to the Korea Institute of Energy Technology Evaluation and Planning (KETEP) for the support grant funded by the Korea government Ministry of Trade, Industry and Energy under the project titled: "Development on localization technologies of export-purposed stationary fuel cell systems", numbered: 20183010032400 and project titled: "Fuel cell safety demonstration linked to hydrogen extractor for public buildings", numbered: 20203040030110.

**Conflicts of Interest:** The authors declare no conflict of interest.

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
