# Peer review of "A New Ammonia Kinetic Model in Ru-Catalyzed Steam-Reforming Reaction Containing N2 in Natural Gas"

_catalysts, doi:10.3390/catal13101380_

Round 1
Reviewer 1 Report
This manuscript presents a kinetic model based on Langmuir-Hinshelwood for ammonia formation in steam reforming reactor and compares with a simplified form of the Temkin-Pyzhev model. This is a novel view on both steam reforming and ammonia formation. However, there are also some important points that need to be addressed.
Comments:
1. In section 2.2, the authors give ammonia formation rate in Equations 14-18. However, Temkin-Pyzhev equation was derived to describe Fe-based catalyst. The kinetic models of Fe-based catalyst and Ru-based catalyst are different due to affinity for H. The authors should give new kinetic model to apply and compare.
2. In Equation 19, why the exponential term of vacant coverage is 2? How to get this conclusion from hydrogen and CO occupying the active sites?
3. In Table 2, what is the data source? The authors should give some reference such as Table 1.
4. For kinetic measurement, the packed bed of catalyst is too long. I wonder whether the influence of internal and external diffusion has been ruled out or not.
5. Some presentation issues. Page 1 Line 35, ‘Steam’ should be lowercase. Page 11 Line 320, H3 or H2? Page 16 Line 439, what is it? Significant figures for all data in the results.
acceptable
Reviewer 2 Report
Dear authors,
The article: A New Ammonia Kinetic Model in Ru catalyzed Steam Reforming Reaction Containing N2 in Natural Gas, is interesting and can be a useful tool to predict the NH3 formation during the steam reforming process. The presentation of the level of confidence for the estimated/experimental values can help in the robustness of the results. Above, the authors have some comments to improve the article:
Line 2 and 3 – The font size looks different
Line 31-33 – Please review the English of the sentence : The method of producing hydrogen by reforming fossil fuels is contact decomposition of fossil fuels on a catalyst by adding water vapor or oxygen.
Line 47 – Please review the English of the sentence: “…damage of not of the…”
Line 64 – replace “implanted” by implemented
Figure 1 – increase the font size in the figure
Line 111 – Delete the r in the sentence“ is rthe …”
Line 118 – eq. 10 , please replace ch4 by CH4, …
Line 143 – indicate the reference
Line 160 – indicate the reference
Figure 3 – in the experiment procedure, in the first box apparently is missing something
Line 302 – please move the table 6 legend to the next page
Table 7 – Can you please show the results with less significant figures
Line 313 – please confirm the legend of figure 5 (is it table 6 ?)
Line 319 – Which is the error confidence level ? 95 % ? 68 % ? Please clarify.
Line 320 – Verify the 4.2 title
Line 322-324 – Review the beginning of this paragraph, it is confused
Line 325 – move the table 9 legend to the next page, please
Line 343 – please correct to Figure 7
Line 342-343 and line 353-354 – the topic is repeated on both paragraphs: “The concentration of NH3 in the outlet gas was calculated by the power law model is compared with experimental data in experiment set 2 and summarized in Figure 6.” and “The comparison between the calculated composition of each gas by the simplified power law model and the measured composition in Experiment set 2 is summarized in Figure 6.”
Line 371 – verify the figure number. Again, which is the level of confidence?
Line 412-414 – The equations should not be repeated, indicate the number of equations is enough
Line 415 – please try to put the table 14 in the same page (without split)
Line 439 – R NH3 (??)
Line 456 & 458 – please, put the 2 in subscript for the N2 and H2
